# Multi-Valent Protein Hybrid Pneumococcal Vaccines: A Strategy for the Next Generation of Vaccines

**DOI:** 10.3390/vaccines9030209

**Published:** 2021-03-02

**Authors:** Ninecia R. Scott, Beth Mann, Elaine I. Tuomanen, Carlos J. Orihuela

**Affiliations:** 1Department of Microbiology, School of Medicine, University of Alabama at Birmingham, Birmingham, AL 35294, USA; nrscott@uab.edu; 2Department of Infectious Diseases, St. Jude Children’s Research Hospital, Memphis, TN 38105, USA; beth.mann@stjude.org (B.M.); elaine.tuomanen@stjude.org (E.I.T.)

**Keywords:** *Streptococcus pneumoniae*, pneumococcal vaccine, hybrid vaccine

## Abstract

*Streptococcus pneumoniae* (*Spn*) is a bacterial pathogen known to colonize the upper respiratory tract and cause serious opportunistic diseases such as pneumonia, bacteremia, sepsis and meningitis. As a consequence, millions of attributable deaths occur annually, especially among infants, the elderly and immunocompromised individuals. Although current vaccines, composed of purified pneumococcal polysaccharide in free form or conjugated to a protein carrier, are widely used and have been demonstrated to be effective in target groups, *Spn* has continued to colonize and cause life-threatening disease in susceptible populations. This lack of broad protection highlights the necessity of improving upon the current “gold standard” pneumococcal vaccines to increase protection both by decreasing colonization and reducing the incidence of sterile-site infections. Over the past century, most of the pneumococcal proteins that play an essential role in colonization and pathogenesis have been identified and characterized. Some of these proteins have the potential to serve as antigens in a multi-valent protein vaccine that confers capsule independent protection. This review seeks to summarize the benefits and limitations of the currently employed vaccine strategies, describes how leading candidate proteins contribute to pneumococcal disease development, and discusses the potential of these proteins as protective antigens—including as a hybrid construct.

## 1. Introduction

*Streptococcus pneumoniae* (*Spn*; the pneumococcus) is an encapsulated Gram-positive bacterium that colonizes the human nasopharynx. It is an opportunistic pathogen that primarily causes disease in infants, the elderly, those who are immunocompromised or have underlying co-morbidities, and individuals with ongoing or recent viral infection of the respiratory tract [1,2,3,4,5]. Once established in the nasopharynx, *Spn* is capable of causing localized inflammation of the upper-respiratory tract resulting in sinusitis, ascending the Eustachian tubes to cause otitis media, being aspirated to cause pneumonia, and, once in the lower respiratory tract, escaping into the bloodstream to cause bacteremia/sepsis and disseminated organ damage including meningitis (Figure 1). Although the incidence rates of *Spn* disease following colonization are overall quite low [6], such a large number of individuals are colonized that the total disease burden is staggering. It has been estimated that the pneumococcus is responsible for ~300,000 deaths in children per year and as much as 1.5 million deaths annually worldwide [7,8]. For these reasons the World Health Organization has designated *Spn* as a priority pathogen.

### Pneumococcal Capsule

Capsular polysaccharide is a principal and well characterized virulence determinant of *Spn*. Composed of repeating oligosaccharide units, at this point in time, 100 biochemically and serologically distinct forms of capsule (i.e., serotypes) are known to exist [9]. Capsule has multiple roles in regard to the biology and pathogenesis of *Spn* [10]. The negative charge of capsule electrostatically repels glycans that are part of mucus, helping to prevent bacterial entrapment and expulsion from the respiratory tract [11]. Capsule is hydrophilic in nature and this confers protection against desiccation during transmission on fomites [12]. Perhaps in its most characterized role, capsule inhibits phagocytosis by immune cells [13]. Capsule does so by inhibiting complement deposition and blocking interactions of receptors on phagocytes, e.g., Fc receptor, with opsonic host proteins bound to the bacterial cell wall or its surface proteins [13]. In similar fashion, capsule has been shown to downmodulate the inflammatory response of immune and non-immune cells by preventing the engagement of Toll-like receptors with PAMPs (pathogen associated molecular patterns) present on the bacterial surface and thereby, dampening *MyD88* downstream signaling and inflammatory cytokine production [14]. As result of its ability to block opsonophagocytosis, *Spn* requires the capsule to survive in the bloodstream, and with extremely rare exception, almost all invasive isolates of *Spn* are encapsulated [15,16].

Since capsule covers the surface of the pneumococcus, antibodies against capsule are highly opsonic, albeit only to *Spn* that produce the capsule type to which the antibody was generated [17,18]. In the bloodstream, where pneumococci must be encapsulated to avoid clearance by immune cells, anti-capsular antibodies are therefore highly protective against invasive disease caused by strains that carry the corresponding serotype. It is for this reason vaccines against *Spn* are currently designed to elicit antibodies against the serotypes most often associated with severe human disease.

## 2. The Rationale for Capsule-Based Vaccines and Their Success

Due to the considerable morbidity and mortality associated with pneumococcal disease, vaccine-based efforts to prevent disease have been ongoing since 1911 [19,20]. Initial vaccines were whole cell-based, involving immunization with heat-killed bacteria belonging to serotype 1, which afflicted mine workers in South Africa [20]. Ultimately, work by MacLeod et al. demonstrated that immunization with capsular polysaccharide conferred protection against disease, and thereafter, vaccine formulations shifted towards the use of purified capsular polysaccharide [21]. Notably and since each serotype is biochemically and antigenically distinct, comprehensive capsule-based immunization coverage against *Spn* would require immunization with the majority of the 100 serotypes that currently exist—which is far too numerous to be feasible. Instead, the vaccines that are currently used against *Spn* are composed of purified capsular polysaccharide from a limited number of serotypes most commonly responsible for human disease [22]. Currently, there are two types of vaccines containing capsular polysaccharides that are approved by most licensing agencies: one composed of 23 purified capsules (PPSV) and, the other composed of 7, 10, or 13 purified capsules conjugated to a protein carrier (PCV) [22,23].

### 2.1. Pneumococcal Polysaccharide Vaccine: PPSV23

PPSV23 was licensed by the U.S. Federal Drug Administration in 1983. It consists of capsules from serotypes: 1, 2, 3, 4, 5, 6B, 7F, 8, 9N, 9V, 10A, 11A, 12F, 14, 15B, 17F, 18C, 19A, 19F, 20, 22F, 23F and 33F [22]. The serotypes incorporated into this vaccine accounted for 60–70% of *Spn* serotypes that caused invasive pneumococcal disease specifically in developed countries [19]. PPSV23 has been shown to be up to 65% effective against invasive pneumococcal disease but does not have demonstrable protection against colonization or pneumonia [19,24]. As a result of the former, it does not negatively impact transmission nor promote herd immunity. Conventionally, polysaccharides are poor stimulators of the adaptive immune system. *Spn* capsular polysaccharides are no different and this is the greatest limitation of PPSV. Given that capsule is a carbohydrate, it is not presented to T-cells in an major histocompatibility complex (MHC) class II dependent manner by antigen-presenting cells (i.e., capsules are a T-cell independent antigens). The consequence of this is that children under the age of 2 years, one of the highest risk groups, fail to elicit an effective protective immunological response to PPSV [25]. In adults, the absence of CD4^+^ cell generated immunity results in the lack of a memory response, thus immunity wanes within 5 years.

### 2.2. Polysaccharide Conjugate Vaccines: PCV7,10, and 13

In order to protect young children and influenced by the success of the polysaccharide conjugate vaccine against *Haemophilus influenzae* type B (HiB) licensed in 1985, polysaccharide conjugate vaccines (PCV) for *Spn* were developed [26]. The fundamental difference between PCV and PPSV is that the polysaccharide capsule is conjugated to a protein carrier in PCVs. The attachment to a protein alters antigen processing within the cells such that the polysaccharide is processed and presented with a fragment of the protein by MHC II and this results in a protective T-cell dependent antibody response against the capsule type [19]. In contrast to PPSV, PCVs protect children under the age of 2 years and elicit long-lasting immunity. The first PCV, PCV7, which covered seven serotypes (4, 6B, 9V, 14, 18C, 19F and 23F) was licensed in the U.S. in 2000 for use in children and dramatically reduced all pneumonia-based hospitalizations by 39–52% [19,22,27,28,29]. A 2007 study in Spain demonstrated that PCV7 was 31% effective against pneumonia and reduced pneumococcal invasive disease by 88% [30]. Strikingly, the efficacy of PCV7 in children under the age of 2 years was 80–90% [31]. Importantly, the robust immunogenicity of PCV7 resulted in sufficiently high levels of IgG antibody against capsule in salivary mucosal secretions to prevent colonization of pneumococci from the nasopharynx of children, the reservoir of *Spn*. This, in turn, reduced transmission and led to a decline in disease in unvaccinated adults, a phenomenon known as herd immunity [32]. In 2010, PCV10 and PCV13 vaccines were introduced to expand coverage to 10 (PCV7 and 1, 5, and 7F) and 13 serotypes (PCV10 and serotypes 3, 6A,19A) [19,22]. These next generation products retain the advantages of conjugate vaccines and are the standard of efficacy upon which all other vaccines will be judged.

Adults over the age of 65 years are recommended to receive pneumococcal vaccination and either the 23-valent capsule based non-conjugate vaccine or PCV13 can be used at the choice of the medical provider. The clinical target of vaccination is pneumococcal pneumonia and, in both cases, the protective immune response is serotype specific. For both vaccines, the overall rates of invasive disease and bacteremic pneumonia in elderly adults were decreased by 50–75% but there was no protection against community acquired pneumonia. The poor activity of capsule-based vaccines of all types in preventing respiratory infection is a major shortcoming that is important for newer vaccines to address [33].

## 3. Limitations of Polysaccharide-Based Vaccines 

Since there are 100 *Spn* serotypes, a 13-valent or 23-valent vaccine targets only a small subset of possible clinical strains. While there will be more complex PCVs in the future, there is a physical limit to the number of serotypes that can be formulated into one vaccine and concerns exist that serotype expansion will reduce the response to each individual capsule type due to in host competition for specific T-cells capable of recognizing the protein component of the vaccine [34]. While it would enhance overall protection, creation of geographically specific combinations of capsule into separate vaccines would increase costs and vaccinated populations would remain subject to serotype replacement and the other limitations described below. The development of a pneumococcal vaccine that is not solely capsule dependent can potentially offer progress past escape from vaccine coverage and provide protection for a broader susceptible population.

### 3.1. Serotype Replacement

Serotype replacement is a process whereby non-vaccine serotypes emerge and move into the niche previously occupied by the vaccine-covered serotypes. It had been hoped that since the 93 non-vaccine serotypes were not as frequent a cause of infection as those covered by PCV7, that they would be less virulent and remain less frequent. This proved inaccurate and the genetic content of emerging serotypes showed no loss of virulence traits [35]. The most prominent example of serotype replacement occurred within four years after PCV7 was introduced. During 2000 to 2004, the yearly incidence for serotype 19A climbed from 2.5/100,000 to 7.8/100,000 [36]. To counter this threat, expanded coverage PCV13, including serotype 19A, was deployed and vaccine efficacy rebounded to 80-82% [31]. However, serotype 35B infection rates are now increasing [37]. It can be hypothesized that serotype replacement will occur each time a new polysaccharide conjugate is developed; thus making it imperative to have a vaccine strategy that circumvents vaccine escape [22].

### 3.2. Antibiotic Resistance

Antibiotic resistance is a relentless concern as pneumococci are naturally competent for DNA exchange and readily acquire resistance to beta lactams, macrolides, fluoroquinolones and vancomycin. This dangerously compromises treatment of the most severe infections, particularly meningitis. The deployment of vaccines has slowed the spread of antibiotic resistance as shown by the decrease in pneumococcal infections by strains resistant to any antibiotic from 7/100,000 to 3/100,000 after PCV7 was introduced in 2000 [38]. As resistance tends to occur in specific serotypes, inclusion of these in vaccine formulations has made a major impact on eliminating resistance. For example, PCV7 successfully targeted over 80% of infections by resistant strains. Beginning in 2010, serotype 19F emerged to cause 30% of resistant infections and addition of this serotype to PCV13 significantly decreased resistant infection. However, just as with serotype replacement, adding capsule types to increasingly complex vaccine formulations is not a long term solution to controlling the threat of antibiotic resistance.

### 3.3. Capsule Shedding and Phase Variation

The success of capsule based vaccines against invasive disease is unfortunately counterbalanced by weak efficacy at mucosal sites, such as for pneumonia and otitis media. Like many other successful bacterial pathogens, *Spn* undergoes phase variation during which surface components, such as capsule, shift stochastically from high to low levels of expression. The frequency of specific phase-variants in distinct anatomical sites in turn being the result of anatomical-site specific selective pressures in the host environment [39]. Bacteria on the mucosal surface have been shown to be less encapsulated than blood-isolated counterparts, thereby exposing surface adhesins, and thus antibodies directed at capsule are at a disadvantage in promoting opsonophagocytosis at the mucosal level. In addition to this genetic adaptation to a niche, a newly recognized phenomenon of capsule shedding also decreases the efficacy of anti-capsular antibodies [40]. Upon entering the lung, *Spn* encounters antimicrobial peptides and as a defense, the bacteria shed most of their capsule over several hours. This floods the pulmonary milieu with detached capsule and simultaneously reveals surface proteins that interact with host cells promoting *Spn* invasion. Antibodies to capsule are thus consumed by this diversion and rendered ineffective.

### 3.4. Serotype 3

Serotype 3 is a particular problem and merits discussion. Notably, the incidence of disease caused by serotype 3 isolates has not declined despite its inclusion in the 13-valent PCV vaccine. Serotype 3 produces profuse quantities of capsule, resulting in a “wet” or mucoid appearance when grown on agar plates. Unlike the vast majority of other serotypes, serotype 3 capsule is not covalently attached to the bacterial surface and therefore is constantly released into the milieu. Similar to shedding of covalently attached capsule by LytA, unbound serotype 3 capsule effectively binds capsular antibody at a distance, away from the bacteria, and therefore negates opsonization [41]. Choi et al. assessed the serotype specific protective effects of PCV-13. It was shown that 0.2 μL of serotype 3 bacterial culture supernatant was sufficient to overwhelm the protective effect of the vaccine-induced antibody in passively immunized mice, a value 100 times more effective at neutralizing anti-capsular antibody than serotype 4 [42]. It was estimated that eight times more antibody is needed to be generated by PCV in order to induce protection against serotype 3 invasive disease [42,43]. Notably disease caused by serotype 3 *Spn* is particularly severe including complications of pneumonia and adverse cardiac events being independently associated with this serotype [44,45,46].

### 3.5. The Elderly Are Highly Susceptible to Pneumococcal Disease

Individuals 65 years and older are particularly susceptible to severe pneumococcal disease and the relative prevalence of specific serotypes differs from that in children [2]. Among adults, *Spn* is responsible for ~37% of all community acquired pneumonia [47], with a stunning >84% of all *Spn* associated mortality occurring among the elderly [48]. The reasons for this increased susceptibility are multifaceted and include structural changes to the respiratory tract, increased frequency of underlying co-morbidities, and changes to both the innate and adaptive immune system that include general declines of protective humoral memory [2]. In adults, as well as children, viral infection, such as influenza, primes the respiratory system for a secondary bacterial infection, most often *Spn* [49]. It is notable that considerable documentation exists supporting the notion that immunization of seniors with PCV23 offers some protection against secondary pneumonia, but that protection is further increased when influenza vaccine is administered together [50]. Altogether, the elderly are at greater risk for opportunistic infection with a much broader range of pneumococcal serotypes than those which cause disease among younger adults as result of a generally less robust immune system [51,52]. A vaccine that confers protection to the elderly must therefore be efficacious against more than the 13 serotypes currently in PCV13.

## 4. Advantages of Multiprotein-Based Vaccines

Although the current capsule-based pneumococcal vaccines have saved millions, there is clearly room for improvement [43]. Several important considerations point to advantages of a vaccine comprised of multiple proteins [53]. Pneumococci infect many body sites and use organ specific virulence determinants to cause disease [54]. A given protein may have different efficacy protecting the lung, heart or brain. Prevention of colonization may involve yet other proteins and recent evidence suggests that preventing transmission is not equivalent to interrupting colonization [55,56]. This opens the concept of multicomponent vaccines that may have different protective functions to achieve broad anti-*Spn* immunity. Since capsule-based vaccines are optimized only to induce opsonic antibodies, the abilities of vaccines to protect by other mechanisms are not currently being incorporated into vaccine design strategies. For example, antibodies that aggregate bacteria for more effective clearance, neutralize toxins, or block *Spn* adherence, invasion, etc. can be induced across serotypes by incorporating proteins into vaccines. This blind spot suggests current designs are missing opportunities to increase the efficacy of existing and future vaccines [57].

The concept that targeting multiple proteins with many functions could lead to broader and potentially superior protection is supported by the mechanism proposed to underpin the broad naturally occurring immunity that develops in young adults. This protection arises by repeated infections, including asymptomatic colonization, experienced throughout childhood and is dependent on the adaptive immune response targeting many *Spn* antigens (e.g., surface proteins) [58]. Rather than being simply serotype specific, immunity in young adults is a result of an adaptive immune response against a variety of conserved bacterial proteins [53]. Importantly, given the sequence variability in pneumococcal surface proteins, it seems that no one protein antigen is likely to be sufficiently conserved to confer protection on its own.

## 5. Leading Candidate Proteins

Leading candidate proteins and their location on the pneumococcal surface are shown in Figure 2. Numerous studies with immunized animals show that antibodies against a combination of pneumococcal proteins confers superior protection compared to single antigens [55,59,60,61,62,63]. Key criteria for choosing optimal proteins include: extent of conservation across *Spn* (and absence in other non-pathogenic streptococci), high in vivo expression across relevant anatomical sites [54], and sufficient surface expression and accessibility suitable for recognition by antibody [64]. Below we discuss the candidate proteins that are being investigated as part of multi-valent protein vaccines either as a mixture or presented as a single recombinant hybrid protein.

### 5.1. Pneumolysin

Pneumolysin is a cholesterol-dependent pore-forming toxin produced by nearly all *Spn* [65]. It is produced as a 53kDa soluble monomer [66]. Pneumolysin does not have an N-terminal signal peptide and is thus localized within the cytoplasm. Its release is dependent on death and lysis of the bacteria [67,68,69]. Upon binding to the host cell membrane, pneumolysin oligomerizes forming a 30–50 subunit pre-pore complex, that subsequently undergoes a conformational shift, becomes hydrophobic, and inserts into the membrane to create a 350–450 Å pore [70,71]. Pore-formation by pneumolysin results in host cell ion-dysregulation which, at low levels, can cause cellular dysfunction, including inhibition of ciliary beating on bronchial epithelial cells, and at higher doses trigger cell death via apoptosis and necroptosis [72,73]. Pneumolysin can kill virtually any host cell. Pneumolysin is one of *Spn*’s most potent virulence determinants, and on its own, following intratracheal instillation into a rat lung, was capable of recapitulating the lung injury observed during pneumococcal pneumonia. Pneumolysin has also been shown to induce cochlear damage and hearing loss and aid in immune system avoidance in animal models [66,74,75]. Recently, it has been classified as an important mediator of cardiomyocyte death during disseminated infection [73,76].

From a vaccine perspective, pneumolysin is highly conserved and antigenic. Human studies have shown that exposed individuals generate high antibody titers against pneumolysin [77,78]. Moreover, pneumolysin cytotoxicity can be effectively neutralized by antibody raised against a toxoid version that fails to oligomerize [79]. However, pneumolysin is not present on the bacterial surface in meaningful concentrations and thus antibody against pneumolysin is neutralizing but not opsonic.

### 5.2. Pneumococcal Surface Protein A (PspA)

PspA is a member of the pneumococcal choline binding protein family [80]. A C-terminal choline binding domain allows these proteins to non-covalently attach to phosphorylcholine residues found on the teichoic acid of the cell wall (Figure 2). PspA has multiple roles including conferring resistance to lactoferrin killing [81], as well as sterically blocking recognition of phosphorylcholine residues on the bacterial surface that target the bacteria for clearance by the host innate immune component C-reactive protein and complement [82].

As a vaccine candidate, the expression of *pspA* is constitutive and among those genes with the highest level of expression across five distinct anatomical sites: the nasopharynx, lungs, blood, heart, and kidneys of infected mice by three distinct strains of *Spn* [54]. Thus, it is a valid protective antigen candidate for antibody-mediated opsonization. In numerous animal studies of immunization followed by direct *Spn* challenge, antibodies against PspA were sufficient to confer protection [83,84]. The N-terminal domain of PspA is composed of numerous alpha-helices and has considerable variability, sufficient to divide the protein into distinct serological-based clades [85,86]. Protective antibodies can be clade specific or directed to the conserved proline-rich domain found in the majority of PspA versions. Passive protection against pneumococcal sepsis in mice was also seen using antibodies generated from a human clinical safety trial [84,87].

### 5.3. Choline Binding Protein A (CbpA/PspC)

Another vital choline binding protein is CbpA [80]. CbpA, also known as PspC, has multiple roles in pathogenesis, including functioning as an adhesin/invasin, modulating chemokine responses, and disrupting the alternative complement cascade by binding to Factor H [88,89,90,91]. Similar to PspA, CbpA has a polymorphic alpha-helical domain at its N-terminus. However, two domains are broadly conserved and suitable as antigens for a vaccine. The (NEEK) domain binds to host laminin receptor (LR) during bacterial translocation across vascular endothelial cells and the YPT domain binds the polymeric immunoglobulin receptor (pIgR) on mucosal epithelial cells in the upper respiratory tract [88,89,92] (Figure 2 and Figure 3). These interactions induce bacterial uptake into cells via receptor-mediated endocytosis and allow the bacteria to cross to the basolateral side of endothelial or epithelial barriers, respectively. This includes crossing the blood-brain barrier to cause meningitis and trafficking into the heart to cause myocardial injury [59,80,89,93,94].

Similar to pneumolysin and PspA, antibodies against CbpA have been demonstrated to confer protection in animal models. Alone and in combination, the conserved epitopes of CbpA generate antibody-mediated and cell-mediated protection at mucosal sites, during invasive disease, and particularly show promise in protecting the lung, heart and brain [63,89,94,95,96]. A unique advantage of CbpA as a vaccine antigen is its ability to generate cross protective antibodies to proteins from meningococcus and *Haemophilus influenzae* that also target LR to promote invasion of the brain during meningitis [89,94].

### 5.4. Pneumococcal Choline Binding Protein A: PcpA

PcpA, which is unfortunately eponymous to but distinguishable from CbpA, is a 79kDa protein with a C-terminal choline-binding domain and several leucine-rich repeats in its N-terminal [97]. It has been demonstrated to be under the control of a manganese-dependent regulator, PsaR [98]. Thus, it is actively expressed in the blood and lungs and repressed in secretions, where the concentration of manganese is low [98,99,100]. PcpA plays a role in bacterial adherence to human pharyngeal and lung epithelial cells [101]. Mice infected with PcpA mutants had decreased lung bacterial burden indicating that this protein is important for promoting pneumonia [98]. Using signature-tagged mutagenesis, PcpA mutants showed decreased virulence during mouse pneumonia and sepsis [102]. Mice immunized with PcpA and infected with *Spn* showed demonstrably lower lung bacterial burdens in a pneumonia model and increased survival in a sepsis model [103,104].

### 5.5. Histidine Triad Protein D: PhtD

PhtD belongs to the pneumococcal histidine triad family known by the HxxHxH motif that may bind metal or nucleosides [60,105,106]. These proteins help to inhibit C3 deposition on *Spn* [107]. Specific PhtD antibodies endow protection against sepsis in a serotype and mouse strain dependent manner [106]. Notably, antibodies against PhtD protect mice infected with serotypes 3, 4, 6A, and 6B, but not serotypes 1 and 5 [61,106]. Acute and convalescent sera collected from human infants and adults demonstrated that PhtD antibodies are generated during natural otitis media and bacteremia [106].

## 6. Why Immunization with More Than One Protein Is Better. A Potential Role for Hybrid Antigens within the Conjugate Vaccine

### 6.1. PcpA/PhtD/Pneumolysin

Given the extraordinary success of PCVs in protecting against invasive disease, it is unlikely that capsule-based vaccines will be replaced. Similarly, there does not appear to be a single master pneumococcal protein that would sufficiently expand protection if used in place of CRM197 in conjugate vaccines. Rather, it is more likely that the increasing need for vaccines against non-PCV serotypes will be addressed by multi-protein vaccines used in parallel with PCVs. These protein combinations have several important capabilities not found in PCVs. They can target beyond opsonophagocytosis to different processes in disease (colonization, transmission, attachment/invasion, cell death) and extend protection beyond PCVs to multiple organs and compartments (upper and lower respiratory tract, heart, and brain). Along such lines, the use of recombinant hybrid proteins that elicit protective antibody against more than one pneumococcal protein would be highly favored as it would enhance vaccine efficacy while limiting production costs.

A trivalent vaccine that included PcpA, PhtD and the pneumolysin toxoid PlyD1 (Figure 3A), was extensively tested in mouse models and showed protection from serotypes 6A and 3-induced pneumonia [108] and infant mouse sepsis [109]. Promising results in models against pneumonia and upper respiratory infection, known deficits of PCVs, resulted in enthusiasm for assessing safety and efficacy in human clinical trials. In several Phase I randomized clinical trials, this triad was uniformly safe and antigenic [94,110,111,112,113,114,115]. Unfortunately, analysis for protection against carriage or otitis media showed no benefit [116,117,118].

### 6.2. Pneumolysin/CbpA Hybrid (YLN) 

YLN (YPT-L460D-NEEK) is a single hybrid protein created by fusing the DNA sequences for two domains of *cbpA* (YPT, pIgR binding domain and NEEK, Laminin receptor binding domain) to a gene for a toxoid of pneumolysin (L460D) (Figure 3B) [63,92,108,109,111,112,113,114,115]. The L460D substitution disrupts the cholesterol recognition motif, thereby strongly reducing cytotoxicity [119]. The vaccine induces antibodies to all three components, including neutralizing wild type toxin and blocking adherence to epithelia and endothelia in invasive disease. In a sepsis mouse model, the mortality of mice vaccinated with YLN was significantly decreased against several pneumococcal strains and this protective phenotype extended to organ specific bacterial burdens as well [63]. Unlike with PCV, the YLN vaccine was demonstrated to be protective against otitis media caused by a non-encapsulated strain [113]. Antibodies against CbpA have been demonstrated to also recognize the LR binding proteins of meningococcus and *H. influenzae* [94] and thus provide cross protection for meningitis and otitis for three frequent pathogens in these sites [88,92,120]. YLN rabbit antiserum given to mice was effective in a passive protection model against sepsis, meningitis, pneumonia and acute otitis media [89]. YLN is the only vaccine thus far to protect against formation of *Spn* cardiac microlesions that lead to cardiac mortality following pneumonia [72,121].

### 6.3. Integrated Approaches

Importantly, one option being considered is that newer versions of the PCV vaccines integrate the use of one or more conserved pneumococcal proteins. Capsular polysaccharide from the serotypes being added to the expanded versions of PCV could be conjugated to an individual or combination of pneumococcal proteins alternatively conjugated to a hybrid protein such as YLN, instead of the currently used carrier protein. Such a vaccine formulation would have the advantage of conferring strong protection against invasive disease caused by the most problematic serotypes, yet also eliciting broad protection against all other pneumococci. Antibody elicited by the inclusion of a hybrid protein versus a single antigen has the advantage of expanded protection. The ability of antibodies against proteins to confer these protective effects would be complementary to the opsonization occurring as result of the antibody generated to capsule and therefore could take advantage of the non-opsonic properties of antibody as discussed above.

## 7. Live and Whole Cell Vaccines as Alternatives

The concept of immunization with multiple proteins can also be achieved by the older strategy of using live or killed intact bacteria. This approach has lost favor as it may result in presentation of unwanted toxic components and technological and genetic advances have allowed isolation of purified proteins. However, these advancements do not diminish the ability of intact cells to invoke serotype-independent immunity. If more advanced strategies fail in the clinic, this approach may be a backup to development of protein vaccines. As expected, intact bacteria, whether alive or heat-killed, induce robust antibodies to many proteins.

At least four live vaccines have been studied using several methods of attenuation such as replacing the pneumolysin gene with a toxoid or deleting genes needed for bacterial metabolism in vivo [56]. Should these vaccines establish a long enough residence time upon inoculation in the nasopharynx, this approach has advantages in presenting antigens to the mucosal surface directly at this entry site. Concerns regarding the feasibility of live vaccines are that the individuals who most benefit by vaccination against *Spn* are those that are the relatively immunocompromised, e.g., very young children, elderly, and individuals with genetic disorders such as sickle-cell disease. The possibility of an attenuated vaccine strain regaining virulence through genetic recombination is also to be considered. Thus, a live vaccine has inherent risks that might not make it tractable.

Two killed unencapsulated whole cell vaccines have been in development. One bears a pneumolysin toxoid gene and a deletion in the major autolysin and the other is a gamma irradiated intact bacterium [122,123]. Both raise antibodies and T cells to a broad array of antigens and are protective in animal models of sepsis and carriage [124]. The former has been reported to be safe and immunogenic in Phase I human trials [125]. There is a significant cost advantage to this simple approach and if side effects prove to be minimal, it may be an important addition to vaccination in the underdeveloped world.

## 8. Conclusions

Pneumococcus has been recognized as a leading cause of morbidity and mortality for over a century. As a result, polysaccharide-based vaccines (PPSV and PCV) were pioneered against this infection to massive success. Although currently licensed PCV is often thought of as the “gold standard”, serotype replacement and other limitations demand a new, potentially added approach–in particular, serious consideration is being given to a protein-based vaccine that confers broad serotype-independent protection. As discussed, a protein-based vaccine has the potential to provide increased respiratory tract protection, prevent invasion of specific organs, neutralize pneumolysin cytotoxicity, provide cross protection against additional childhood pathogens and mitigate adverse complications arising from disseminated bacterial infection. The hope is that promising protein-based vaccines will mount protective and long-lasting memory that is conserved across pneumococcal serotypes.

## Figures and Tables

**Figure 1 vaccines-09-00209-f001:**
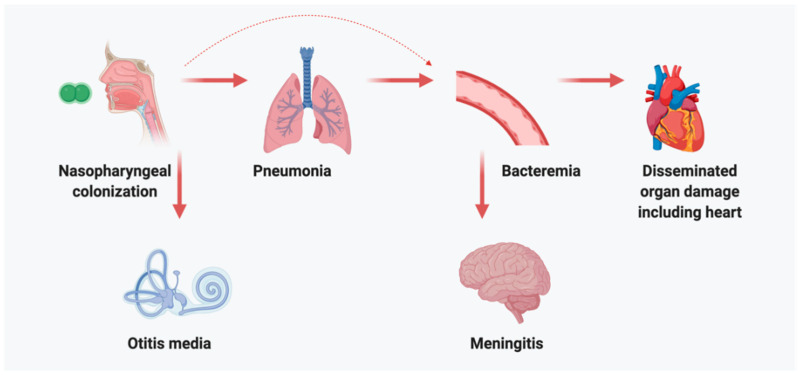
Pneumococcal pathogenesis. *Streptococcus pneumoniae* (*Spn*) pathogenesis begins with successful colonization of the nasopharynx, where it typically resides asymptomatically. Given the opportunity, particularly in those who have recently experienced viral infection or are immunocompromised, *Spn* can cause localized disease in the middle ear (otitis media) and in the lungs (pneumonia). Failure to control these infections, allows *Spn* to escape into the bloodstream (bacteremia) and cause sepsis, systemic disseminated organ damage, which includes the heart, and central nervous system.

**Figure 2 vaccines-09-00209-f002:**
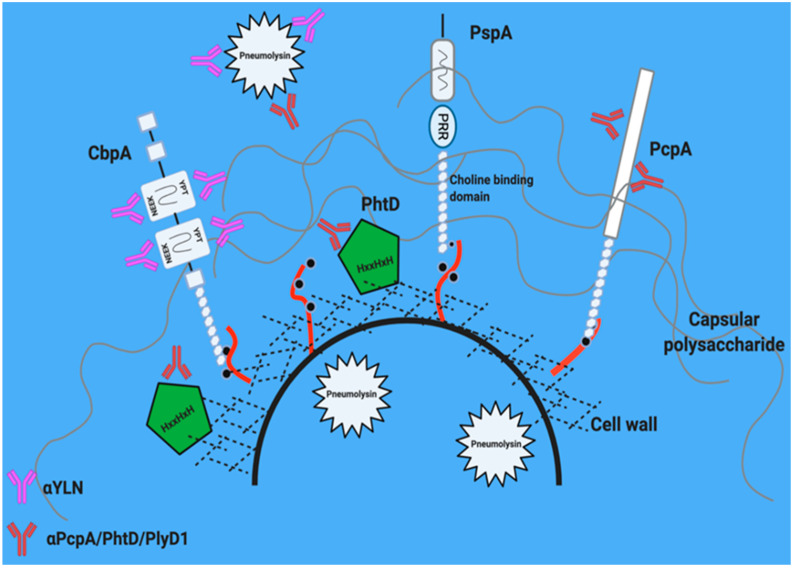
Leading candidate proteins for multi-valent vaccines. There are several conserved bacterial proteins—both on the surface of the cell wall (hatched black lines with red teichoic acids) and present in the extracellular environment of the bacteria. Pneumolysin (white star) is a pore forming toxin that is localized within the cytoplasm, but is released during cell lysis. Pneumococcal surface protein A (PspA), choline binding protein (CbpA), pneumococcal choline binding protein (PcpA) and histidine triad protein D (PhtD) are surface proteins that have domains conserved across serotypes and contribute to pneumococcal pathogenesis. Antibodies (pink and red Y shapes) targeting each of these proteins have been shown to convey protection to animals against experimental disease. Thus, they have been formulated into vaccines for human clinical trials.

**Figure 3 vaccines-09-00209-f003:**
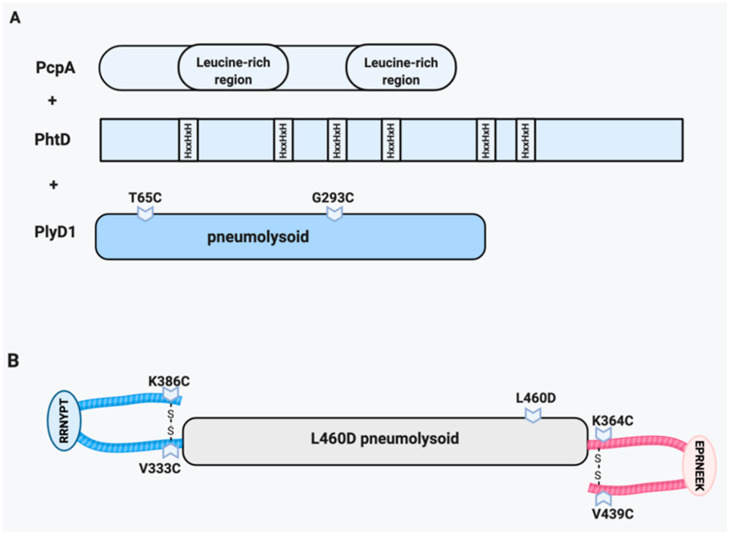
Multivalent protein formulations. (**A**) The PcpA, PhtD, PlyD1 multivalent vaccine consisting of recombinant PcpA and PhtD along with the PlyD1 pneumolysin toxoid. (**B**) YLN hybrid construct consisting of the YPT and NEEK conserved domains of CbpA fused with the N- and C-teriminus, respectively, of the inactive L460D pneumolysin toxoid. 6.1. PcpA/PhtD/pneumolysin toxoid multivalent vaccine.

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
