# Peer review of "Multi-Valent Protein Hybrid Pneumococcal Vaccines: A Strategy for the Next Generation of Vaccines"

_vaccines, 2021, doi:10.3390/vaccines9030209_

Round 1

Reviewer 1 Report

The review article with the manuscript (ID): Vaccine-1108210, submitted by Scott et. al., entitled “Multi-valent protein hybrid pneumococcal vaccines: a strategy for the next generation of vaccines” described nicely the concept that the protein-based vaccines might offer a promising protective efficacy and long-lasting memory, which would be conserved within the highly divergent pneumococcal serotypes in the field. However, this novel approach has needs to be randomized clinical trials to insight into the potential efficacy against pneumococcal infections. The article has an outstanding merits and interests in this field, and well-written and also presented nicely as well. However, authors are suggested to provide a bit more description under the part of discussion, which could improve the quality of the article.

At page 10, line-398, …….it could be ‘whether a live or killed,….”. Please revise the line.

Author Response

We thank Reviewer 1 for the highly positive comments. Below we address the two minor concerns raised by the reviewer.

  1. provide a bit more description under the part of discussion, which would improve the quality of the article.

Importantly, Reviewer 3 indicated that “it would be preferable to have a 'conclusion' instead of discussion as discussions have been made in the previous sections”. We reached out to the editor for clarification on whether to expand our discussion section or rename the section as conclusion. The editor indicated that a conclusion section was required. Given the thorough discussion throughout each of the sections, and to avoid redundancy, we chose to instead rename the discussion as “conclusion” (ln 466). 

  1. At page 10, line-398, …….it could be ‘whether a live or killed….”. Please revise the line.

We have changed the line to “whether alive or heat-killed” (ln 447).

Reviewer 2 Report

Brief Summary: This paper aims to review current vaccine strategies as well as the potential for new and more effective approaches to vaccines for Streptococcus pneumonia (Spn), a WHO designated priority opportunistic pathogen that can colonize the human nasopharynx, and can cause disease in infants, the elderly, those who are immunocompromised, and those what have recently had viral infection of the respiratory tract. There have been well established vaccines developed and used to prevent infections by this pathogen that are composed of purified pneumococcal capsular polysaccharide in a free form or conjugated to a protein carrier. While these vaccines are shown to be fairly effective in both protecting the host as well as helping to create herd immunity, there are some noticeable gaps in immunity to Spn even with these vaccine types, and there is concern that these vaccines don’t provide enough immunity against certain Spn bacterial serotypes. The proposal to add extra serotypes into these vaccines to generate increased immunity to Spn strains is also a concern/not effective, as it is hypothesized that serotype expansion in vaccines could reduce the immune response to each individual capsule type. The authors of this paper suggest that vaccines for Spn can possibly be improved upon with the introduction of using Multi-valent protein hybrid pneumococcal vaccines either as a replacement or in addition to well established vaccines. By using a multiprotein-based vaccine for Spn the authors hypothesize that protein-based vaccines can establish additional avenues of immunity by stimulating antibodies that are more effective at bacterial clearance, neutralizing toxins, blocking Spn adherence and invasion, etc. Taking this approach also does not need to be serotype specific, hypothetically leading to a broader range of protection to specific Spn strains. Experiments using animal models have already shown promise with increased immunity to Spn when inoculated with multi-valent protein hybrid pneumococcal vaccines. Overall, the authors of the paper hypothesize that protein-based vaccines may lead to increased immunity to Spn as they are able to stimulate the body to create antibodies that are effective at clearing Spn from the host that is able to reach broader immunity to Spn strains, as it is not serotype-specific.

Broad comments: There were also a couple of points made in the “Why capsule-based vaccines are limited” section (Page 3-4) that could have been expanded upon that would have really highlighted why it is important to research other avenues of immunity to Spn despite the capsular-based vaccine being widely accepted as effective. I also think that it would be helpful to expand a bit more about why it is important that we have access to a vaccine to Spn that is not serotype dependent (i.e. the emergence of new strains that are prominent in infection, broader immunity in general, etc.).

Specific comments: On page 4 lines 136-137 mention that there is a concern that expanding the serotypes included in capsular-based vaccines reduces the response to each individual capsule type. I think that this is a very important point highlighting why targeting specific serotypes can pose risks, but the paper does not explain at all how that works, so a brief sentence or two explaining why that happens/why that has been hypothesized could help highlight why adding additional serotypes to capsular-specific/serotype specific vaccines could pose risks in the future. On page 5 the sentence from line 212-213 mentions that preventing transmission is not equivalent to interrupting colonization. Do serotype/capsular specific vaccines only prevent transmission/don’t prevent colonization? That may have been mentioned and I missed it, but if it wasn’t already established it would be important to highlight that as a weakness of capsular-based vaccines in the manuscript. 

Author Response

We thank Reviewer 2 for the highly positive comments. Below we address the concerns raised by the reviewer.

  1. There were also a couple of points made in the “why capsule based vaccines are limited sections (3-4) that could have been expanded upon that would have really highlighted why it is important to research other avenues of immunity to Spn despite the capsular-based vaccine being widely accepted as effective. I also think that it would be helpful to expand a bit more about why it is important that we have access to a vaccine to Spnthat is not serotype dependent (i.e. the emergence of new strains that are prominent in infection, broader immunity in general, etc.).

We agree reviewer that there are key limitations to current capsule-based vaccines. The entirety of section 3 delineates these limitations, which are serotype replacement (ln 162), antibiotic resistance (ln 176), capsule shedding and phase variation (ln 190), serotype 3 (ln 211), and susceptibility within the elderly (ln 228). Note that the antibiotic resistance section is all new and in response to the reviewers. We have also added the statement “The development of a pneumococcal vaccine that is not solely capsule dependent can potentially offer progress past escape from vaccine coverage and provide protection for a broader susceptible population.” (ln 159-161).

  1. On page 4 lines 136-137 mention that there is a concern that expanding the serotypes included in the capsular-based vaccines reduces the response to each individual capsule type. I think this is a very important point highlighting why targeting specific serotypes can pose risks, but the paper does not explain at all how that works, so a brief sentence or two explaining why that happens/why that has been hypothesized could help highlight why additional serotypes to capsule specific/serotype specific vaccines could pose risks in the future.

We thank the reviewer for raising this important point. We have modified the statement such that it now indicates that reduced vaccine efficacy is possibly due to competition for T-cells that can recognize the protein component of PCV (line 155-156).

  1. On page 5 the sentence from line 212-213 mentions that preventing transmission is not equivalent to interrupting colonization. Do serotype/capsular specific vaccines only prevent transmission/don’t prevent colonization? That may have been mentioned and I missed it, but if it wasn’t already established it would be important to highlight that as a weakness of capsular-based vaccines in the manuscript. 

Lines 127-131 specifically mention that PCV confers sufficient antibody so as to prevent colonization and confer protection by blocking subsequent transmission. In rereading the text, we understand that our statement was indeed confusing. We have modified the statement mentioned by the reviewer (now line 239-240) so as to avoid this and be more concise.

Reviewer 3 Report

  1. Correction in Figure 1 (Line 51): Disseminated organ damage which includes the heart and menigitis. It should be 'meninges'.
  2. In the introduction section, from Line 52-75, the author describes the structure of Streptococcus pneumoniae capsule, its role in pathogenesis, how it interacts with host immune system and host immunity elicited against capsular antigens. I suggest to make it a section with a separate heading (e.g. 2. Pneumococcal capsule).
  3. In section 2.2, the authors have highlighted the effectiveness of polysaccharide conjugate vaccines (PCV) for children however the effectiveness of PCV in elderly adults has not been discussed.
  4. Line 132: The heading may be reformatted such as 'limitations of the polysaccharide based vaccines'.
  5. Line 139: Correct the double space (remain subject)
  6. Line 189: The heading should be 3.4 (and not 3.3)
  7. In relation to the section 4, incorporation of the studies showing that the naturally acquired protection in mouse models against invasive pneumococcal disease depends on antibody to protein antigens rather than capsule can further highlight the protective role of protein antigens and their future use as a vaccine . (Wilson R, Cohen JM, Reglinski M et al. Naturally Acquired Human Immunity to Pneumococcus Is Dependent on Antibody to Protein Antigens).
  8. Line 350: 6.1 PcpA/PhtD/pneumolysin should be in the next line as a sub-heading.
  9. Section 8 (Line 417): It would be preferable to have a 'conclusion' instead of discussion as discussions have been made in the previous sections. 
  10. The rise in antibiotic resistance among non-vaccine serotypes is another important factor to look for alternative strategies. In my opinion, this should be highlighted in the relevant section.

Author Response

We thank Reviewer 3 for their constructive comments. These have been addressed as detailed below.

  1. Correction in Figure 1 (Line 51): Disseminated organ damage which includes the heart and menigitis. It should be 'meninges'.

We have corrected this mistake and changed it to “central nervous system” (ln 50-51).

  1. In the introduction section, from Line 52-75, the author describes the structure of Streptococcus pneumoniae capsule, its role in pathogenesis, how it interacts with host immune system and host immunity elicited against capsular antigens. I suggest to make it a section with a separate heading (g. 2. Pneumococcal capsule).

We have made the change suggested.

  1. In section 2.2, the authors have highlighted the effectiveness of polysaccharide conjugate vaccines (PCV) for children however the effectiveness of PCV in elderly adults has not been discussed.

A paragraph in regard to the effectiveness of PCV in adults over age 65 is now added (line 136-148).

  1. Line 132: The heading may be reformatted such as 'limitations of the polysaccharide based vaccines'.

We have made the change suggested.

  1. Line 139: Correct the double space (remain subject)

We have made the change suggested.

  1. Line 189: The heading should be 3.4 (and not 3.3)

We have made the change suggested.

  1. In relation to the section 4, incorporation of the studies showing that the naturally acquired protection in mouse models against invasive pneumococcal disease depends on antibody to protein antigens rather than capsule can further highlight the protective role of protein antigens and their future use as a vaccine . (Wilson R, Cohen JM, Reglinski M et al. Naturally Acquired Human Immunity to Pneumococcus Is Dependent on Antibody to Protein Antigens).

We agree and statements to this effect and the suggested Wilson et al. reference are on line 255 and 275.

  1. Line 350: 6.1 PcpA/PhtD/pneumolysin should be in the next line as a sub-heading.

We have made the change suggested.

  1. Section 8 (Line 417): It would be preferable to have a 'conclusion' instead of discussion as discussions have been made in the previous sections. 

We have made the change suggested.

  1. The rise in antibiotic resistance among non-vaccine serotypes is another important factor to look for alternative strategies. In my opinion, this should be highlighted in the relevant section.

We noted and have taken the reviewers comment on adding the section on antibiotic resistance (ln 176) to address this.